# Quantitative Analysis of Surface Attached Mortar for Recycled Coarse Aggregate

**DOI:** 10.3390/ma15010257

**Published:** 2021-12-30

**Authors:** Guoying Liu, Qiuyi Li, Jialin Song, Liang Wang, Haibao Liu, Yuanxin Guo, Gongbing Yue

**Affiliations:** 1School of Architectural Engineering, Qingdao Agricultural University, Qingdao 266109, China; 20212204005@stu.qau.edu.cn (G.L.); lqyyxn@163.com (Q.L.); jiangongwl_2019@163.com (L.W.); guoyuanxin@qau.edu.cn (Y.G.); 2Linyi Blue Thai Environmental Protection Technology Co., Ltd., Linyi 276000, China; songjialinooo@163.com; 3School of Civil Engineering, Qingdao University of Technology, Qingdao 266033, China; lhbhxw@163.com

**Keywords:** recycled coarse aggregate, attached mortar, calcination temperature, grinding time, microhardness

## Abstract

Due to the large amount of old hardened cement mortar attached to the surface of aggregate and the internal micro-cracks formed by the crushing process, the water absorption, apparent density, and crushing index of recycled coarse aggregate are still far behind those of natural coarse aggregate. Based on the performance requirements of different qualities of recycled coarse aggregate, the performance differences of recycled coarse aggregate before and after physical strengthening were observed. The results showed that the physical strengthening technique can remove old hardened mortar and micro powder attached to the surface of recycled coarse aggregate by mechanical action, which can effectively improve the quality of recycled coarse aggregate. The optimum calcination temperature of the recycled coarse aggregate was 400 °C and the grinding time was 20 min. The contents of the attached mortar in recycled coarse aggregates of Class I, II, and III were 7.9%, 22.8%, and 39.7%, respectively. The quality of recycled coarse aggregate was closely related to the amount of mortar attached to the surface. The higher the mortar content, the higher the water absorption, lower apparent density, and higher crushing index of the recycled coarse aggregate.

## 1. Introduction

Construction waste treatment and recycling is an important part of the circular economy, and its significance is beyond doubt. Recycled coarse aggregate prepared by crushing and screening construction waste and applied to recycled concrete [1] can consume a large amount of accumulated construction waste, which is conducive to the promotion of a sustainable development strategy. Compared to the natural aggregate, the surface of the recycled coarse aggregate prepared from waste concrete is attached with a large number of hardened cement stones with low strength, high water absorption, and weak combination with aggregate [2], resulting in the existence of multiple interface structures in recycled concrete, resulting in significant deterioration of the mechanical properties and durability of recycled coarse aggregate concrete [3,4]. In order to improve recycled coarse aggregate, researchers worldwide have carried out a large number of experiments on the modification [5,6,7,8] and ratio [9,10] of recycled coarse aggregate, which can be summarized as physical technology, chemical technology, carbonization technology, and nanotechnology.

(1) Physical technology [11,12,13,14] is the basic idea of removing the waste cement paste attached to the recycled coarse aggregate by kneading, heating, particle forming, micro-heating, ultrasonic cleaning, and other processes. (2) Chemical technique [15,16,17,18,19,20] consists of immersing the recycled coarse aggregate in different kinds of chemical grouting and can be mixed with additive gold powder, silica powder, fly ash, and any other fine ore powder or slag. (3) Carbonization technology [21,22,23,24,25] is when newly collected aggregates are put into the atmosphere with high CO_2_ concentration, CaCO_3_ is formed by the reaction of the cement slurry with CO_2_, and CaCO_3_ is deposited in the pores or cracks to improve the performance of recycled coarse aggregate or strengthen recycled coarse aggregate by microbial induced mineralization deposition of calcium carbonate [26]. (4) Nanotechnology [27,28,29] is the use of nano-materials to promote hydration, reactions with cement-based materials, filling pores, and controlling the crystallization process. Compared with other methods, the physical strengthening method is more economical and practical, and more conducive to the promotion and applications of industrialization.

There is a great difference in the thermal expansion coefficient between the aggregate and mortar [30]. Attached mortar and coarse aggregate will produce thermal strain due to temperature change. At the same time as thermal strain, there is thermal stress between the aggregate and slurry, and the old interface of the recycled coarse aggregate surface will be damaged due to the development of thermal stress. With the increase in temperature, the damage will gradually accumulate between the old interface and produce obvious microcracks, which will significantly reduce the bonding force of the old interface. At the same time, the C–S–H gel will also produce large shrinkage after high temperature dehydration [31,32]. Cao Beibei [33] studied the difference in thermal expansion of cement mortar and aggregate at different temperatures. Linear expansion was used to represent the elongation of different materials affected by temperature and it was found that the expansion coefficient of each component would also change with the change in temperature. When the temperature reaches 700 °C, the expansion rate of coarse aggregate is twice that of the linear expansion rate of mortar, and the waste mortar becomes brittle and its hardness value decreases significantly after high-temperature calcination, while the hardness value of the old aggregate does not change significantly. The waste mortar attached to the surface of the recycled coarse aggregate can be easily separated by an external force [34,35,36].

Therefore, based on the performance requirements of the recycled coarse aggregate of different qualities, the performance differences of recycled coarse aggregate before and after physical strengthening, and separated the attached mortar on the surface of recycled coarse aggregate from the old aggregate by using the collision principle of high-speed rotating ball grinding beads were investigated to determine the content of recycled coarse aggregate attached mortar.

## 2. Experiment

### 2.1. Raw Materials

The recycled coarse aggregate was obtained from the waste concrete with a strength grade of C40. The physical strengthening method of particle shaping can effectively improve the quality of recycled coarse aggregate. The basic principle is to strip the attached mortar on the surface of the recycled coarse aggregate by high-speed impact and grinding between the aggregate and equipment and between the aggregate and aggregate. In this way, high-quality recycled coarse aggregate with a low content of attached mortar and good shape morphology can be obtained. The production principle of particle shaping equipment is shown in Figure 1. The equipment can effectively reshape and strengthen the low quality recycled coarse aggregate, pulverize and collect dust, prepare high-quality recycled coarse aggregate and recycled micro powder for recycled concrete, and effectively reduce dust emission.

After jaw crusher crushing screening, a 5–25 mm simple crushing of class I recycled coarse aggregate can be obtained using a physical reinforced plastic device to a part of the class I recycled coarse aggregate physical strengthening treatment after screening, a 5–25 mm physically strengthened class II recycled coarse aggregate can be collected, again when recycled coarse aggregate physically reinforces a class II, one can obtain a high quality of class III recycled coarse aggregate. Recycled coarse aggregate comes from discarded concrete with a strength grade of C40, which uses physical strengthening technology [37] to obtain class I, class II, and class III recycled coarse aggregates. The grading curve of different recycled coarse aggregates is shown in Figure 2.

According to the national standard method (GB/T25177-2010), the performance indexes of recycled coarse aggregate were measured. The influence of physical strengthening technology on the quality and properties of recycled coarse aggregate was analyzed, and the grade of different recycled coarse aggregates was evaluated. The obtained test results are shown in Table 1.

After the physical strengthening process, the mortar and micropowder attached to the surface of the recycled coarse aggregate were removed mechanically, and the grain shape and basic properties of the aggregate were significantly improved, which effectively improved the quality of the recycled coarse aggregate. Among them, the content of micropowder was reduced by 68.4%, the water absorption was reduced by 69.8%, the content of needle flakes was reduced by 80.3%, and the crushing index was reduced to 9.5%, and the apparent density reached 2474 kg/m^3^, fully meeting the standard for Class I recycled coarse aggregate.

### 2.2. Principles and Solutions

#### 2.2.1. Calcining-Grinding Principle

High temperature calcination produces the difference in linear expansion between the aggregate and attached mortar. The large stress difference caused by thermal expansion makes the thermal compatibility of the granite in the recycled coarse aggregate and the old mortar worse, which results in greater damage in the interface transition zone, and the overall structural performance of recycled coarse aggregate is destroyed.

The old interface in the recycled coarse aggregate will produce tiny cracks after calcination at high temperature. The separation efficiency of waste hardened mortar can be improved by using these cracks and loose interfacial transition zone. The heated aggregate was put into the planetary ball mill for ball grinding, and the waste hardened mortar was stripped from the surface of the recycled coarse aggregate through the collision and grinding between materials and materials, and between ball grinding beads and materials. While removing the waste mortar, it also reduces the prominent edges and corners on the surface of the recycled coarse aggregate, so that the aggregate particles tend to be smooth to achieve the effect of ball grinding on the recycled aggregate.

Due to the irregular shape of the recycled coarse aggregate (as shown in Figure 3), many parts of the aggregate (as shown in Figure 4b) cannot touch the grinding beads, and the grinding effect cannot be achieved when using the ball mill beads with a larger particle size. Only the waste mortar (as shown in Figure 4c) wrapped on the protruding part of the original natural aggregate surface can be ground. Therefore, the continuous particle size ball milling beads were used for grinding to completely remove the waste mortar on any part of the original natural aggregate surface, and the accurate-attached content of recycled coarse aggregate can be obtained.

Based on the above discussion, in order to more efficiently separate the mortar attached to the surface of the recycled coarse aggregate, the mortar attached to the surface of recycled coarse aggregate was removed by a calcination-ball milling method. The morphological changes of different quality recycled coarse aggregates at different temperatures were observed, and the optimum calcination temperature and adequate ball milling time were determined based on the removing rate of the sintered-grinding mortar, and the attached mortar amount of different qualities of recycled aggregate mortars was eventually revealed.

#### 2.2.2. Experimental Design

(1)Natural coarse aggregates with the size of 5–25 mm and recycled coarse aggregates of class I, class II, and class III were calcined at different temperatures (100 °C, 200 °C, 300 °C, 400 °C, 500 °C, 600 °C, respectively), placed in a planetary ball mill after cooling. Twenty batches of various coarse aggregates were randomly selected, and the average value was taken after each batch was ground three times;(2)The grinding time was sequentially controlled at 5 min, 10 min, 15 min, 20 min, 25 min, and 30 min. After reaching the specified grinding time, we sieved and weighed the remaining amount M of the aggregate at different grinding times, and calculated the mass loss rate of two adjacent grinding time points;(3)In order to eliminate the loss and influence of the original natural aggregate during the grinding process, the natural aggregate was taken as a control group. Grinding can be terminated when the quality loss rate of various recycled coarse aggregates approaches the same as that of natural aggregates, and the amount of attached mortar removed A is:


(1)
$$A=\frac{M_{\mathrm{natural}}-M_{\mathrm{RCA}}}{M_{\mathrm{sample}}}\times 100\%$$


In the formula, A refers to the amount of attached mortar removed, %; M_RCA_ refers to the mass of recycled coarse aggregate after ball milling, g; M_natural_ refers to the mass of natural coarse aggregate after ball milling, g; and M_sample_ refers to the mass of sample before ball milling, g.

The morphology of the recycled coarse aggregate after high temperature calcination is shown in Figure 5. Observing the appearance of the recycled aggregate after calcination, it can be seen that there were obvious cracks in the old interface and the interface transition zone and the mortar structure was relatively loose, but the appearance of the aggregate had no obvious change. The aggregate morphology after ball milling is shown in Figure 6. After different calcination temperatures and grinding times, the particle size of the recycled aggregate and natural aggregate was close to pebble, and there was basically no mortar attached to the surface of the recycled coarse aggregate.

## 3. Results and Discussions

### 3.1. Interfacial Structure of Recycled Coarse Aggregate

It can be seen from Figure 7 that the old interface between the original aggregate and the attached mortar was relatively clear. Figure 7a shows that the structure of the original aggregate was uniform and compact, without micro cracks, while the attached mortar had a large number of irregular microcracks and holes. The microstructure was relatively loose. There was a large number of irregular micro-cracks in the mortar attached to the surface of the recycled coarse aggregates, which were generated from the interface transition zone and extended to the direction of the mortar matrix. These microcracks are the weak link of recycled coarse aggregate concrete. The attached mortar on the surface of the coarse aggregate was removed and Figure 7b shows that there was a big difference between the microcracks L1 and L2. There were more hydration products inside L1, and there were more fine cracks around and extending to the surroundings. It can be judged that this kind of crack is caused by the original concrete itself and has nothing to do with the crushing of the waste concrete. However, there was no hydration product filling in the L2, and the crack form was single, indicating that this kind of crack was relatively dense in hydration, but there was a lot of surface. The holes were prone to cracks at the edges and extended to the outside. Nucleus crystals formed inside the holes, mainly in the form of floc-like crystals. The image was caused by mechanical damage. Figure 7c shows that due to the long hydration age of the mortar matrix, there were obvious cracks at the old interface, and the hydration products were abundant and the structure was relatively loose at the cracks. The hydration products were mainly flocculent. Enrichment of clumpy C–S–H gel and flake Ca(OH)_2_ led to large crystal particles [38] and high porosity at the interface junction, which was also the main weak link of recycled concrete.

### 3.2. Determination of Calcination Temperature

The natural coarse aggregates were calcined at 100 °C, 200 °C, 300 °C, 400 °C, 500 °C, 600 °C, and normal temperature, respectively. Three samples for each temperature level was selected, and the testing samples were cut into 10 mm slices. These were then subjected to grinding and polishing treatment to meet the requirements of the microhardness test specimens (Figure 8). Six measurement areas were taken for each sample, each measurement area had 16 measurement points, so that the microhardness of each measurement point could be calculated, the six maximum and six minimum values of the 96 microhardness values were removed before taking the average value. The distribution of measuring points is shown in Figure 9.

Calculation of the loss rate of natural aggregate microhardness at different temperatures:(2)$$S=\frac{H_{V}\left(T_{0}\right)-H_{V}\left(T_{1}\right)}{H_{V}\left(T_{0}\right)}\times 100\%\;$$

In the formula, S is the microhardness loss rate, %; H_V_(T_0_) is the normal environmental hardness value, MPa; and H_V_(T_1_) is the hardness value after T_1_ high temperature calcination, MPa.

Figure 10 shows the microhardness loss rate of natural aggregates at different calcination temperatures. The calcination temperature below 100 °C had little effect on the aggregate. At 400 °C, the microhardness decreased to 245 MPa, the hardness loss rate was 14.6%, and the hardness of the aggregate was gradually damaged. When the temperature rose to over 400 °C, the microhardness value of the natural aggregate changed rapidly, and the hardness loss rate increased to 41.2%. After calcining the natural aggregate to 700 °C, the cracks and collapse that appeared around the indentation were obvious in the microhardness test, and the diagonal line was not obvious to read for the data. It can be concluded that excessive temperature will damage the hardness of the aggregates. Therefore, considering the high energy consumption and the damage to the hardness of the coarse aggregate, it was finally determined that the calcination temperature of the recycled coarse aggregate could not be higher than 400 °C.

### 3.3. Quantitative Analysis of Attached Mortar

Due to the different calcination temperature, the grinding time of the attached mortar on the surface of the recycled coarse aggregate was different. After the calcination temperature exceeded 400 °C, the damage to the old aggregate in the recycled coarse aggregate was more serious. Therefore, in order to research the content of the attached mortar on the surface of the aggregate, the temperature was controlled in the range of 100–400 °C, and by adjusting the grinding time, testing, and analyzing the content of the attached mortar, the calcination temperature and grinding time can eventually be determined.

It can be seen from Figure 11 that when the calcination temperature of class I recycled coarse aggregate was below 200 °C, the removal rate of waste mortar was relatively small, and when the calcination temperature reached 300 and 400 °C, the removal of the attached mortar became more obvious. After 20 min of grinding time, the removal rate of the two was roughly equal and tended to be stable, and the contents of the attached mortar were 7.5% and 7.9%, respectively. The class II and III recycled coarse aggregates changed significantly with the calcination temperature. The removing rates after calcination at 400 °C and grinding for 20 min were 22.8% and 39.7%, respectively. Therefore, the calcination-grinding method was used to remove the attached mortar on the surface of the recycled aggregates, the calcination temperature of class I, class II, and class III recycled coarse aggregates should be controlled below 400 °C, and the optimum grinding time was 20 min.

After the low-quality recycled coarse aggregate underwent particle shaping and strengthening treatment, the attached amount of waste mortar was effectively removed under the action of external force. In particular, the high-quality recycled coarse aggregate after the shaping and strengthening treatment contained less waste mortar on the surface, its surface had no edges and corners and the grain shape was relatively smooth, and the content of the mortar obtained after calcination and grinding was only 7.5%. Therefore, it was also proven that the use of the particle shaping physical strengthening method can effectively reduce the waste mortar attached to the surface of the recycled coarse aggregate, and improve the basic performance and quality of the recycled coarse aggregate. With the increase in the calcination temperature, the removal rate of the recycled aggregate gradually increased. This was mainly due to the difference in thermal expansion between the aggregate and mortar under the action of the high temperature of recycled coarse aggregate, which produced an obvious thermal stress concentration at the interface transition zone and produced a large number of microcracks and damage at the interface transition zone and the mortar matrix, which weakened the interface structure performance of the recycled aggregate [39]. Under the action of mechanical external force and self-collision, the separation and recycling of recycled aggregate and old mortar were realized, and the grain shape of the treated recycled aggregate was relatively round.

### 3.4. Correlation between Attached Mortar Content and Technical Index

By fitting the measured test data, the relationship between the attached amount of recycled aggregate mortar and water absorption, apparent density, and crushing index was obtained. It can be seen from Figure 12, Figure 13 and Figure 14 that the quality of the recycled coarse aggregate is closely related to the amount of mortar attached to the surface. The greater the mortar content, the higher the water absorption rate, the lower apparent density, and the higher crushing index of the aggregate. The content of type III RCA mortar was as high as 34.5%. After high temperature calcination and grinding, the water absorption and crushing indexes of RCA mortar decreased by 78.2% and 67.6%, respectively, and the apparent density increased by 294 kg/m^3^. The content of II type RCA attached mortar was 22.8%, the water absorption rate and crushing index increased 65.2% and 57.4%, respectively after high temperature calcination-grinding, and the apparent density decreased 177 kg/m^3^. The basic properties of the II and III type aggregates were obviously lower than that of the high-quality recycled coarse aggregate (I type). This is because the attached mortar on the surface of class II and class III RCA formed internal damage such as micro voids and cracks in the attached mortar and the transition zone of the old interface in the crushing process, which led to the degradation of its performance.

After calcination and grinding, the water absorption, apparent density, and crushing index of class I recycled coarse aggregate showed relatively little change, and was similar to the natural aggregate. This was because after two-time physical strengthening, the content of attached mortar on the surface of class I recycled coarse aggregate decreased obviously, but there was still a small amount of hardened cement mortar. In contrast, the properties of recycled coarse aggregate of class II and class III were significantly affected by the content of the attached mortar. Figure 13 shows that the apparent density of the natural coarse aggregate increased by 2%, which was due to the decrease in surface edges and angles of the natural aggregate after high-temperature calcination and grinding, which reduced its porosity and slightly increased its apparent density, but high temperature made its crushing index value increase by 7%.

It can be concluded from Figure 12, Figure 13 and Figure 14 that there was a good linear relationship between the content of attached mortar on the surface of the recycled coarse aggregate and its water absorption, apparent density, and crushing index. The correlation coefficient R^2^ was 0.9724, 0.9567, and 0.9889, respectively, which is very consistent with the measured value. After high temperature calcining and high-speed grinding, the water absorption, apparent density, and crushing indexes of the recycled coarse aggregate were basically the same as those of the natural aggregate, which indicates that the high-temperature calcination-grinding method can effectively remove the attached mortar on the surface of the recycled coarse aggregate. For the recycled coarse aggregate used in practical application engineering, only the water absorption rate, apparent density, and crushing index of the recycled coarse aggregate can be detected. Using the established linear relationship, the content of waste mortar attached to the surface of different recycled coarse aggregates can be accurately measured, and the internal defects of recycled coarse aggregate can also be quantitatively reflected.

## 4. Conclusions

In this study, based on the performance requirements of different recycled coarse aggregates, the performance of recycled coarse aggregates through the process of physical strengthening methods were comprehensively compared and analyzed, and the content of attached mortar of the recycled coarse aggregate was quantitatively analyzed by the calcination-grinding method, where the following results were obtained.

(1)The physical strengthening method can effectively reduce the content of attached mortar to the surface of the recycled coarse aggregate, improving the performance of recycled aggregate. Through the secondary physical strengthening process, the water absorption rate of recycled coarse aggregate was reduced to 2.2% from 7.3%, the needle flake content was only 1.2%, and the crushing index was only 3.1% higher than that of the natural coarse aggregate, which fully meets the standard of the class I recycled aggregate.(2)The calcining-grinding method can effectively separate the recycled aggregate and attached old cement mortar. The calcination temperature below 100 °C had no great influence on the aggregate. When the temperature was 400 °C, the microhardness decreased to 245 MPa, and the hardness loss rate was 14.6%. The hardness of the aggregate had been damaged. Combined with the quality change of different-class recycled coarse aggregate and the microhardness loss of the aggregate under high temperature, the appropriate calcination temperature for the class I, II, and III recycled coarse aggregate was determined as 400 °C and the optimum grinding time was 20 min.(3)The measured attached mortar contents of classs I, II, and III recycled coarse aggregates were 7.9%, 22.8%, and 39.7%, respectively, by calcining-grinding technology. Meanwhile, the water absorption, apparent density, and crush index had a good linear relationship, and the established linear relationship could accurate measure the content of the RCA adhesive mortar on the surface, which can directly reflect the internal defects of RCA.

## Figures and Tables

**Figure 1 materials-15-00257-f001:**
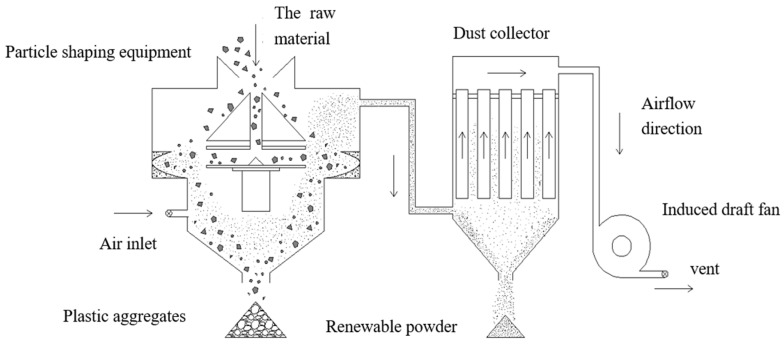
Schematic diagram of the particle shaping equipment.

**Figure 2 materials-15-00257-f002:**
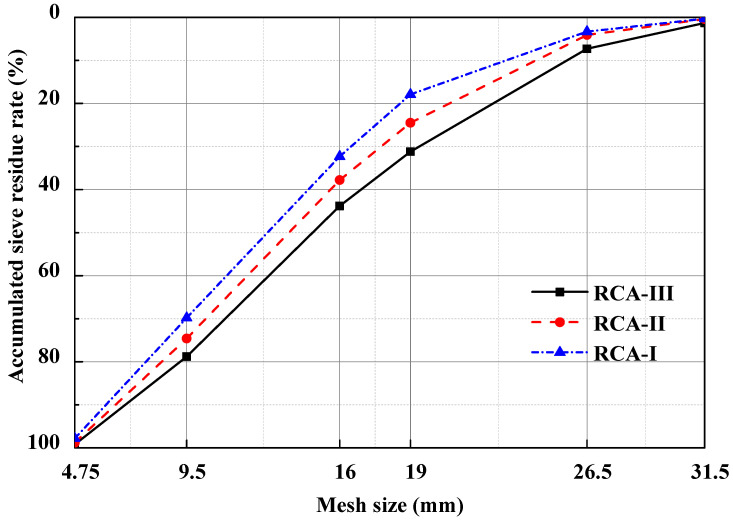
Gradation curve of the different recycled coarse aggregates.

**Figure 3 materials-15-00257-f003:**
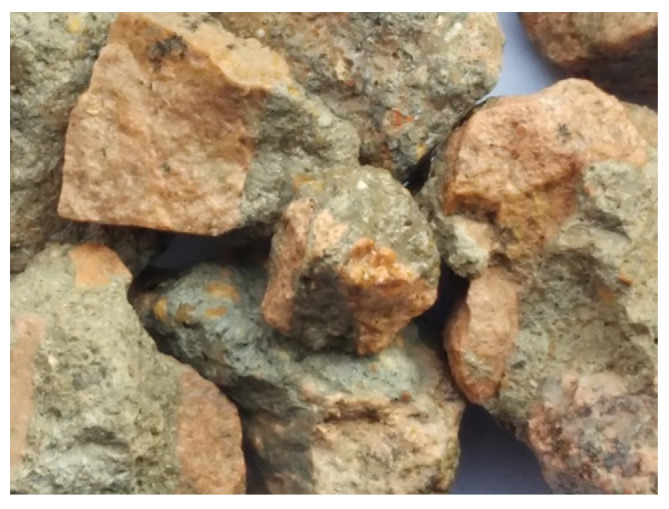
Appearance of the recycled coarse aggregate and attached mortar.

**Figure 4 materials-15-00257-f004:**
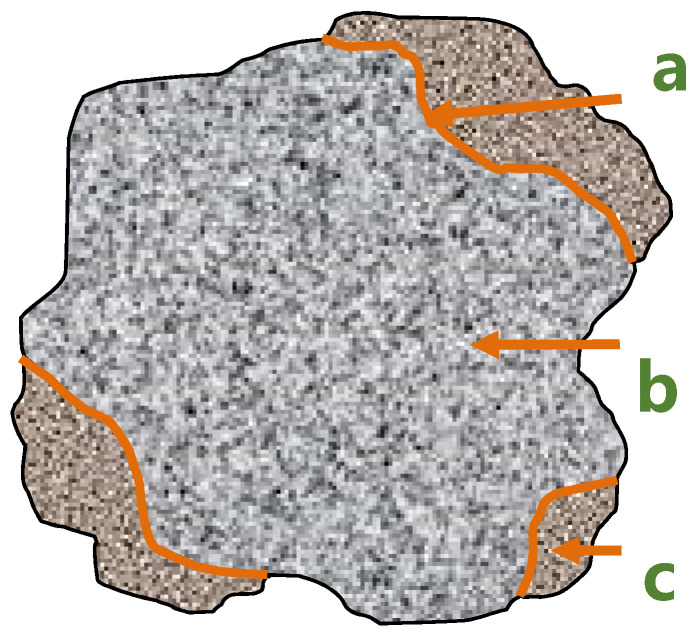
Appearance diagram of the recycled coarse aggregate: (**a**) The old interface; (**b**) Original natural aggregate; (**c**) Waste mortar.

**Figure 5 materials-15-00257-f005:**
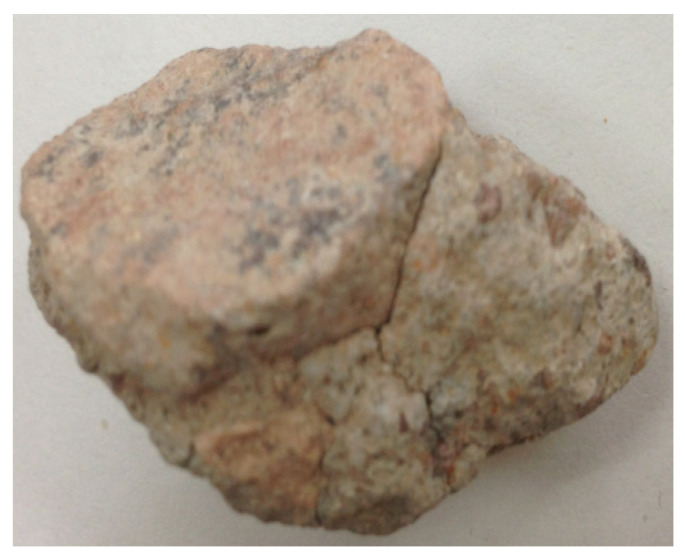
Aggregate morphology after calcination.

**Figure 6 materials-15-00257-f006:**
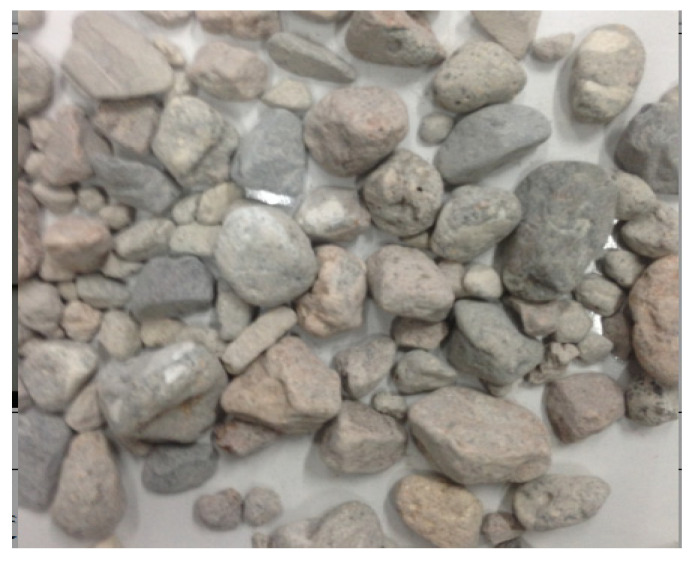
Aggregate morphology after ball milling.

**Figure 7 materials-15-00257-f007:**
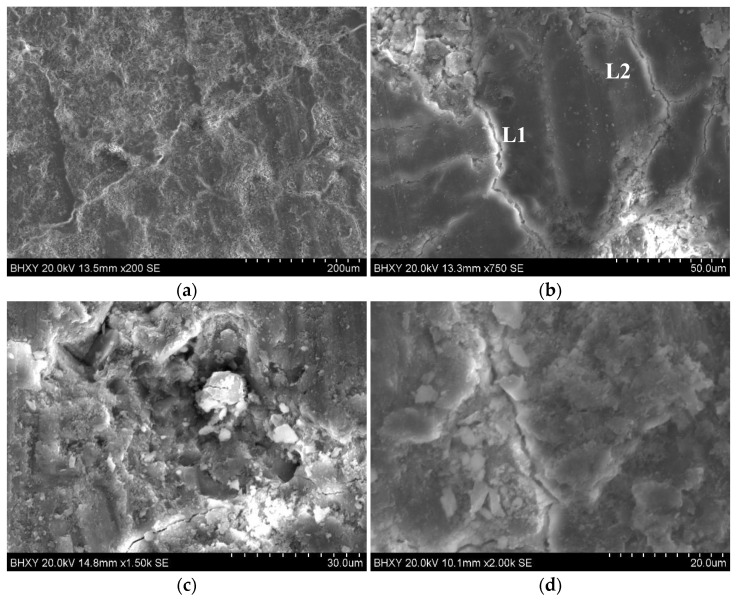
The SEM images of the transition zone of the old interface of recycled coarse aggregate: (**a**) 200 times; (**b**) 750 times; (**c**) 1500 times; (**d**) 2000 times.

**Figure 8 materials-15-00257-f008:**
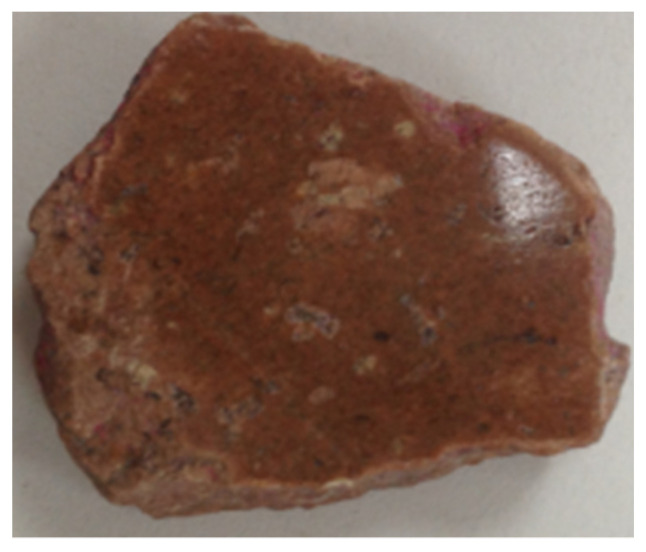
Appearance of the polished aggregate.

**Figure 9 materials-15-00257-f009:**
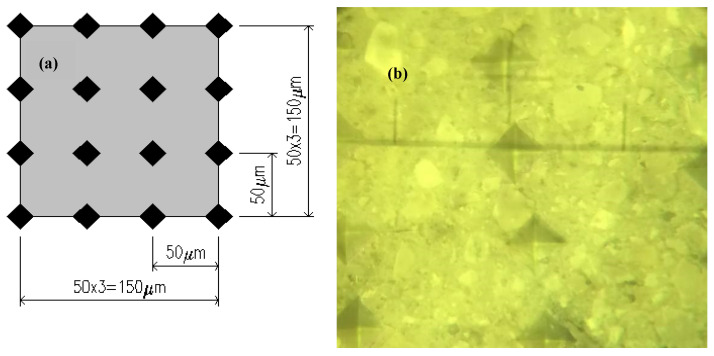
(**a**) Design of the microhardness lattice. (**b**) Microhardness indentation area.

**Figure 10 materials-15-00257-f010:**
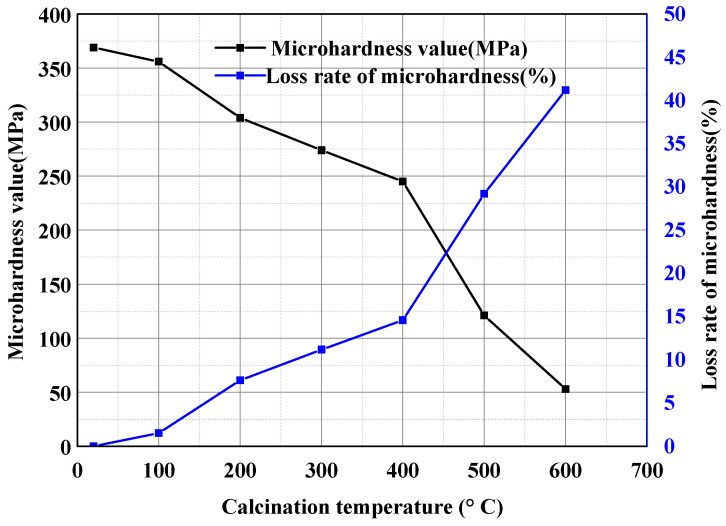
The microhardness and loss rate of natural aggregates at different calcination temperatures.

**Figure 11 materials-15-00257-f011:**
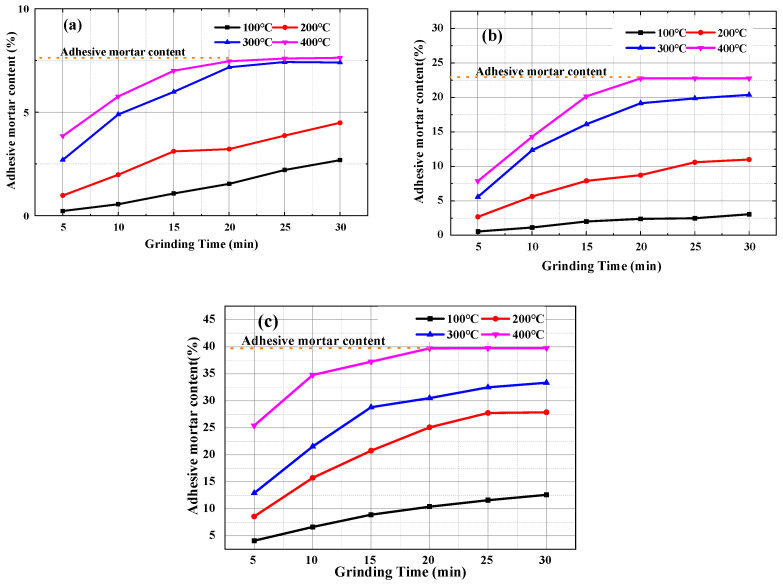
Contents of mortar attached to different quality recycled coarse aggregates. (**a**) Class I recycled coarse aggregate; (**b**) Class II recycled coarse aggregate; (**c**) Class III recycled coarse aggregate.

**Figure 12 materials-15-00257-f012:**
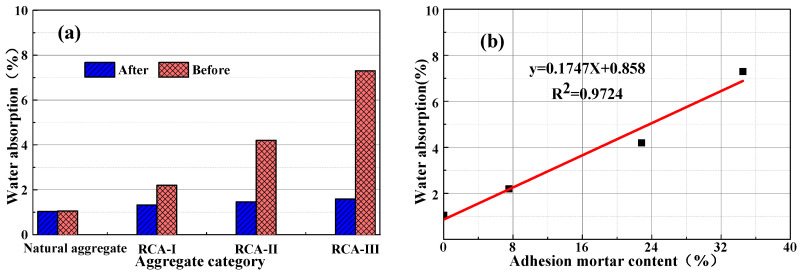
(**a**) Influence of calcination and grinding on the water absorption of recycled aggregate. (**b**) The relationship between the content of attached mortar and water absorption.

**Figure 13 materials-15-00257-f013:**
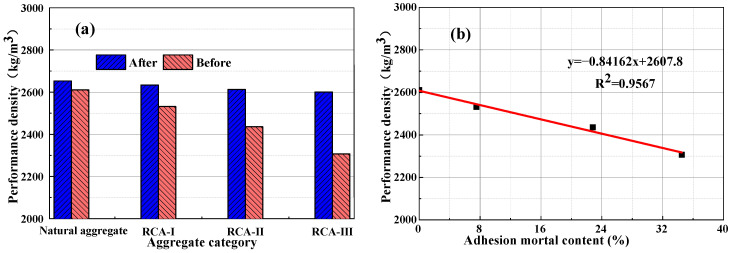
(**a**) The effect of calcination and grinding on the apparent density of recycled aggregate. (**b**) The relationship between the content of attached mortar and the apparent density.

**Figure 14 materials-15-00257-f014:**
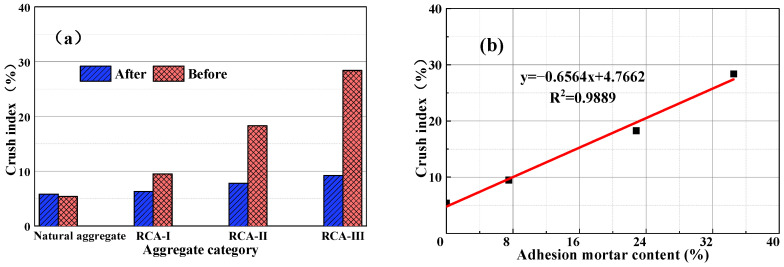
(**a**) Influence of calcination and grinding on the crushing index of recycled aggregate. (**b**) Relationship between the content of attached mortar and crushing index.

**Table 1 materials-15-00257-t001:** Basic performance and evaluation of different recycled coarse aggregates.

Items	Standard Requirements			Recycled Coarse Aggregate Performance		
	Class I	Class II	Class III	A	B	C
Particle grading	Qualified	Qualified	Qualified	Qualified	Qualified	Qualified
Micropowder content/%	<1.0	<2.0	<3.0	1.9	1.2	0.6
Water absorption rate/%	<3.0	<5.0	<8.0	7.3	4.2	2.2
Needle flake particle content/%	<10.0			6.1	3.3	1.2
Sundries content/%	<1.0			1.2	0.5	0.1
Robustness/%	<5.0	<10.0	<15.0	14.3	7.6	3.8
Crushed index/%	<12.0	<20.0	<30.0	28.4	18.3	9.5
Apparent density/(kg/m^3^)	>2450	>2350	>2250	2307	2436	2474
Porosity/%	<47.0	<50.0	<53.0	45.0	44.0	41.0
Alkali aggregate reaction	Qualified			Qualified	Qualified	Qualified

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
