# Peer review of "Quantitative Analysis of Surface Attached Mortar for Recycled Coarse Aggregate"

_materials, 2021, doi:10.3390/ma15010257_

Round 1

Reviewer 1 Report

This is an interesting study with broader implications for civil engineering materials. This study advances research in recycled aggregate concrete manufacturing. This study requires minor improvements before its consideration for publication.

*Revise the abstract. Include more details on the methodology and main findings. First sentence of paragraph is bit difficult to comprehend and therefore should be improved.

*Second sentence of abstract should also be revised to present statements in correct grammar and sentence structure.

*In abstract briefly explain how physical strengthening improved the properties of recycled aggregates in terms of quantity.

*Mention the age of C40 concrete waste.

*Overall manuscript is good

*The paper should answer the environmental significance of the physical strengthening technique over other ecofriendly substitutes. These following studies mentioned ecofriendly and effective solutions. Include them in introduction.

A step towards durable, ductile and sustainable concrete: Simultaneous incorporation of recycled aggregates, glass fiber and fly ash, B Ali, LA Qureshi, SHA Shah, SU Rehman, I Hussain, M Iqbal Construction and Building Materials 251, 118980 (2020).

Mechanical and durability performance of recycled aggregate concrete incorporating low calcium bentonite B Masood, A Elahi, S Barbhuiya, B Ali Construction and Building Materials 237, 117760 (2020).

Combined effects of supplementary cementitious materials (silica fume, GGBS, fly ash and rice husk ash) and steel fiber on the hardened properties of recycled aggregate concrete LA Qureshi, B Ali, A Ali Construction and Building Materials 263, 120636 (2020).

Author Response

Thank you very much for your affirmation of the content of my manuscript.  I have revised all your suggestions.  And upload the modified version to the attachment.  

1. Modified the abstract to make the language smoother and the content more specific.  

 2. Modified the introduction and added excellent literature to make the content more perfect  

 Thank you again for your review of my paper!

Reviewer 2 Report

I revised the manuscript of  Gouying Liu, Qiuyi Li, Liang Wang, Haibo Liu, Yuanxin Guo, Gongbong Yue

The content of the paper is interesting and well organized. The manuscript’s main question addressed is the evaluation of the content of recycled coarse aggregate attached mortar, which is a relevant and interesting point of view, considering that the use of recycled aggregate for the production of concretes is one of the main principles of the Circular Economy considerations.

The manuscript concerns a usual subject area, the topic is well known between the scientific community, but the authors in their paper focus their attention on a specific aspect of the problem that is the influence that the mortar, attached to the particles of aggregate, have when the aggregate has been used as materials for new concrete.

I do not feel enough qualified for evaluating the English level of the manuscript but in my opinion, it is well organized and easy to read. However, the authors should pay attention to some typo errors in the text. For example, the list of authors finished with and *, it is not clear if they forgot to write the name of another author, or they type an extra "and".

The same for the references, the authors should uniform the reference list following the journal instruction.

I appreciate the decision of the authors to discuss the results of their work in the same paragraph of the results, anyway, I think the conclusion paragraph needs to have some introductive lines to introduce the list of the 3-point summarizing the final results of the work.

In conclusion, as concerns the content of the manuscript, I think that it is interesting with good scientific soundness.

Are the conclusions consistent with the evidence and arguments presented? Do they address the main question posed?

Author Response

Thank you very much for your affirmation of the content of my manuscript. I have revised all the opinions you put forward.  And I uploaded the modified version to the attachment.

1. I examined my manuscript carefully again, corrected some spelling mistakes, and revised the references in accordance with the format required by the magazine.  

2. In the conclusion paragraph, several quotations are added to help better align the statement with the evidence presented.  

Thank you again for your review of my paper !

Wish you a happy life!  

Reviewer 3 Report

Manuscript entitled "Quantitative analysis of surface attached mortar for recycled coarse aggregate" is very interesting scientific work describing the problem of usability of recycled coarse aggregate. 

The introduction is well organised however there is lack of some informations. The Authors explain what are the issues of using such aggregate and what are the ideas how solve different problems connected with the recycle coarse aggregates but on the other hand the Authors have just pointed out the solution withouth deeper explenation of using different methods. Maybe presenting some pros and cons of each will clarify why this topic is important.

Moreover there is an issue with clearly understand the novelty and the motivation of the conducted research because they are not clearly specified in the introduction. 

The methodology is understandable but there might be some issues with repeating similar studies. The photographs of conducted research and deeper explanaition of the methodology will clarify this thing. 

Very positive aspect is the way how the results are presented. However the conclusions might be imrpoved and more information can be stated if the Authors compare the results with other similar works. It also may clarify why this method is succesfull or not. 

Conclusion need to be rewritten because now it is just a section with 3 statemets according to the research not to the general knowledge. 

Taking above into consideration, the article is worth publishing however after the major revision. 

Author Response

Thank you very much for your affirmation of the content of my manuscript.  I have revised the manuscript.  And upload the modified version to the attachment.  

1. In this paper, the advantages of physical reinforcement method are described and the reasons for choosing this method are expounded.  

2. Some data comparison was added in the article to make the content more convincing.  

3. Modified the conclusion paragraph by adding appropriate descriptions to make it less general  

Thank you again for your review of my paper!  

Round 2

Reviewer 3 Report

I accept this version